# Fabrication of Anatase TiO_2_/PVDF Composite Membrane for Oil-in-Water Emulsion Separation and Dye Photocatalytic Degradation

**DOI:** 10.3390/membranes13030364

**Published:** 2023-03-22

**Authors:** Chengcai Li, Hewei Yu, Biao Huang, Guojin Liu, Yuhai Guo, Hailin Zhu, Bin Yu

**Affiliations:** 1Zhejiang Provincial Key Laboratory of Fiber Materials and Manufacturing Technology, Zhejiang Sci-Tech University, Hangzhou 310018, China; 2Zhejiang Provincial Innovation Center of Advanced Textile Technology, Shaoxing 312000, China; 3Zhejiang Sci-Tech University Huzhou Research Institute Co., Ltd., Huzhou 313000, China

**Keywords:** membrane, TiO_2_, emulsion separation, photocatalytic degradation

## Abstract

At present, the types of pollutants in wastewater are more and more complicated, however, the multifunctional membrane materials are in short supply. To prepare a membrane with both high efficient oil-in-water emulsion separation performance and photocatalytic degradation performance of organic dyes, the bifunctional separation membrane was successfully prepared by electrostatic spinning technology of PVDF/PEMA and in situ deposition of anatase TiO_2_ nanoparticles containing Ti^3+^ and oxygen vacancies (O_v_). The prepared composite membrane has excellent hydrophilic properties (WCA = 15.65), underwater oleophobic properties (UOCA = 156.69), and photocatalytic performance. These composite membranes have high separation efficiency and outstanding anti-fouling performance, the oil removal efficiency reaches 98.95%, and the flux recovery rate (FRR) reaches 99.19% for soybean oil-in-water emulsion. In addition, the composite membrane has outstanding photocatalytic degradation performance, with 97% and 90.2% degradation of RhB and AG-25 under UV conditions, respectively. Several oil-in-water separation and dye degradation experiments show that the PVDF composite membrane has excellent reuse performance. Based on these results, this study opens new avenues for the preparation of multifunctional reusable membranes for the water treatment field.

## 1. Introduction

In recent years, the rapid development of industrialization has brought critical environmental pollution issues. Industries such as textile, leather, printing and paper-making, etc., tend to create large amounts of organic dyes wastewater [1,2,3,4], while metallurgical, chemical, new energy sources, and pharmaceutical, etc., tend to drain large amounts of oily wastewater [5,6,7], which is very harmful to human health. To effectively solve the water pollution problem, photocatalytic technology and membrane separation technology have become a research hot spot. 

Photocatalysis is one of the most common methods to remove organic dyes, with the advantages of environmentally friendly operation and high photo-degradation efficiency [8,9]. However, most of the catalysts used in photocatalytic technology are powders, which are difficult to recycle and reuse. Membrane separation technology has low energy consumption, high efficiency, simple operation, no secondary pollution, etc. [10,11,12]. Loading the photocatalyst into the membrane can effectively solve the problems of difficult recycling and reuse of the catalyst [13,14]. However, organic dye wastewater often contains a large amount of oil contamination and surfactants. Oil contamination and surfactants will adsorb on the membrane surface, which will reduce the active sites in the membrane and lead to the weakening of the catalytic activity of the membrane. Therefore, it is of great significance to develop new separation membranes with intellectualized, multifunctional, and controllable characteristics for diverse wastewater systems and complicated environments.

The bifunctional composite membranes are a new development trend in the field of membrane materials. For example, a bifunctional composite membrane can not only perform high-efficiency oil-in-water emulsion separation, but also effectively degrade organic dyes in wastewater. In order to endow excellent degradation ability without sacrificing separation performance, the strategy of combining photocatalytically active nanoparticles with hydrophilic membrane materials to prepare bifunctional composite membrane materials has attracted extensive attention in water treatment [8,15]. Xiang et al. loaded MIL-53(Fe) nanoparticles into polyacrylonitrile membrane via electrospinning to achieve highly-efficient oil-in-water emulsion separation and photocatalytic organic dye degradation [16]. Xie at al. loaded NH_2_-MIL-88B(Fe) nanoparticles into PVDF membranes via a one-step non-solvent induced phase inversion method to achieve highly-efficient oil-in-water emulsion separation and photocatalytic organic dye degradation [17]. However, these nanoparticles are not only unevenly dispersed in the polymer, but also the surface of nanoparticles is easily encapsulated by the polymer, weakening the degradation ability and separation performance of the membrane. At the same time, the types of dyes degraded by these membranes are relatively single (such as rhodamine B, methyl orange, and other cationic dyes). Therefore, a novel and simple preparation method is urgently needed to obtain high-performance bifunctional composite membranes.

Titanium dioxide (TiO_2_), a semiconductor material, mainly has three main crystal forms—anatase, rutile, and brookite—and is currently widely used in the field of photocatalytic degradation [18,19,20]. Among them, due to the wider band gap of anatase crystalline TiO_2_, it has the highest photocatalytic activity and is the most widely used [21]. In the preparation of nanosized TiO_2_, its crystal structure can be effectively controlled by controlling some conditions. For example, Yin et al. successfully prepared anatase TiO_2_ with clear cube-like geometry by using hydrogen fluoride and hydrogen chloride as synergistic catalysts [22]. Wang et al. synthesized uniformly sized cubic anatase TiO_2_ nanoparticles by changing titanium hydroxide precipitates into their corresponding ethanol gels, followed by drying in supercritical ethanol (7.0 MPa, 270 °C) and calcination at high temperature (AS-preparation); the anatase TiO_2_ nanocrystallites have highly-efficient photocatalytic methyl orange degradation [8].

In addition, several studies have found that surface defects and oxygen vacancies (O_v_) present on the catalyst surface can facilitate charge transfer, thereby enhancing the photocatalytic activity of photocatalysts [23,24,25,26]. For example, Ji et al. successfully prepared TiO_2_ nanoparticles with Ti^3+^ and O_v_ by in situ growth method and vacuum freeze-drying, the nanoparticles have higher photocatalytic activity compared to commercial TiO_2_ [27]. Qi et al. prepared TiO_2_ with Ti^3+^ and O_v_ through vacuum heating activation treatment, these TiO_2_ shows high photoactivity and photosensitivity [28]. Therefore, inspired by the above, it is speculated that the carrier separation efficiency and photocatalytic activity can be effectively improved by synthesizing anatase-type TiO_2_ with Ti^3+^ and O_v_.

In this work, poly(ethylene-alt-maleic anhydride) (PEMA) was used as the wetting agent, PEMA and Polyvinylidene fluoride (PVDF) were blended to prepare the base membrane by electrospinning method, and then the maleic anhydride in the fiber membrane was hydrolyzed, so as to prepare the carboxy-rich hydrophilic PVDF composite membrane. Last, in situ deposition of anatase TiO_2_ with Ti^3+^ and O_v_ on the hydrophilic PVDF composite membrane surface by a hydrothermal reaction method for oil-in-water emulsion separation and photocatalyze the degradation organic dyes. The preparation process of the bifunctional composite membrane is shown in Figure 1. Abundant carboxyl group and TiO_2_ in membrane improve the permeation flux and catalytic degradation performance of the hybrid membrane. The surface morphology, chemical composition, wettability and water flux, mechanical properties, oil-in-water emulsion separation properties, photocatalytic degradation properties of different dyes, and reusable properties of the composite membranes were investigated by SEM, XRD, FTIR, XPS, WCA, etc. The separation mechanism and photocatalytic degradation mechanism of the bifunctional composite membrane were analyzed. The obtained TiO_2_/PVDF composite membrane showed several distinct advantages: (1) the membrane achieves remarkable underwater superoleophobicity property; (2) the membrane can effectively separate oil/water emulsion and good anti-fouling ability; (3) the membrane has high photocatalytic degradation ability for both anionic and cationic dyes. The work show that the composite membrane has great application potential in the field of wastewater treatment.

## 2. Materials and Methods

### 2.1. Materials

PVDF (Solvay 6010, Mn = 500,000) was obtained from Solvay S.A. (New York, NY, USA). PEMA (Mn = 50,000) was purchased from Vertellus (New York, NY, USA). Tetrabutyl titanate (TBOT), Rhodamine B (RhB, Appendix A), Acid Green 25 (AG-25, Appendix A) were purchased from Shanghai Macklin Biochemical Co., Ltd. (Shanghai, China). N-N-Dimethylacetamide (DMAc), Acetone, HCl, H_2_SO_4_, Tween-80, and dichloroethane (EDC) were obtained from Hangzhou Gaojing Chemicals Co., Ltd. (Hangzhou, China).

### 2.2. Preparation of Carboxy-Rich PVDF Composite Membrane

The carboxy-rich PVDF composite membrane was obtained by a simple blending electrospinning technique. Firstly, 2 g PVDF and PEMA (PEMA is 0 g, 0.1 g, 0.2 g, 0.3 g, 0.4 g, 0.5 g, 0.6 g) blended powder was dissolved in 20 mL DMAc/Acetone (*v*/*v* = 3:2) solvent to obtain a uniform spinning solution. The spinning process was carried out in a closed spinning machine at room temperature and a relative humidity of about 50%. The feeding speed and voltage were set as 0.5 mL/h and 15 kV, respectively, and a collection device covered by an aluminum foil was placed 15 cm away from the needle. After the spinning, the obtained membrane was dried in a vacuum oven at 50 °C to remove residual solvent and then immersed in 0.1 mol/L HCl solution for 1 h before being dried 60 °C for 3 h. Membranes prepared with different PEMA contents were named M0/1/2/3/4/5/6. The hydrophilicity and tensile properties of the membrane were tested to select the membrane prepared with the best PEMA content, and follow-up experiments were carried out.

### 2.3. Preparation of Bifunctional Composite Membrane Loading Anatase TiO_2_ with Ti^3+^ and O_v_

In situ deposition of TiO_2_ on the membrane surface by a hydrothermal reaction method [29]: 1 mL TBOT was dripped into 30 mL of 2 mol/L H_2_SO_4_ solution and stirred for 30 min, and the PVDF composite membranes was immersed in the above reaction solution. After reacting at 120 °C for 1, 2, 3, and 4 h, respectively, named as membranes M5-TiO_2_ (1), M5-TiO_2_ (2), M5-TiO_2_ (3), M5-TiO_2_ (4), respectively.

### 2.4. Emulsion Separation

A composite membrane with an effective area of 13.85 cm^2^ was pre-moistened with deionized water and then fixed on the filter cartridge of the dead-end filtration device under a transmembrane pressure of 0.1 bar. The permeation flux of the membrane was calculated by Equation (1):(1)$$J=\left(\frac{V}{A\times \Delta t}\right)$$
where *J* (L·m^−2^·h^−1^) is the preparation flux, *A* (m^2^) is the effective membrane area, *V* (L) is the volume of permeation over a time interval Δt (h).

The oil-in-water emulsion was prepared as follows: 1000 g of water was mixed with 1 g of soybean oil, and then 0.1 g of Tween-80 was added. Lastly, the mixed liquor was stirred at 4000 rpm for 10 min. The maximum UV absorption wavelength of soybean oil was tested by UV spectrophotometer (Appendix A). The oil-in-water emulsion separation experiment was operated under a transmembrane pressure of 0.1 bar. The particle size of oil droplets and oil content in water before and after filtration was tested by dynamic light scatterer (DLS) (LB 550, HORIBA) and UV spectrophotometer at 220 nm, respectively. The separation efficiency was calculated according to the oil rejection efficiency of Equation (2) [30,31]
(2)$$R=\left(1-\frac{C_{p}}{C_{0}}\right)\times 100\%$$
where *R* (%) is oil rejection efficiency, *C*_0_ is the oil concentration of emulsion, and *C_p_* is the oil concentration of filtrate.

To evaluate the anti-fouling performance of the composite membrane, the flux recovery rate (FRR) was calculated by the following Formula (3).
(3)$$FRR=\left(1-\frac{J_{p}}{J_{0}}\right)\times 100\%$$
where *J*_0_ is the pure water flux, *J_P_* is the pure water flux after oil-in-water emulsion separation and cleaning.

### 2.5. Photocatalytic Degradation Performance of Membranes

The photocatalytic activities of the composite membrane were analysed by RhB (initial concentration 5 mg/L, pH = 7) and AG-25 (initial concentration 50 mg/L, pH = 3) in the dark and under UV irradiation, respectively. A schematic of the photoreactor equipment is illustrated in Figure 2. The 30 mg sample was immersed in 150 mL dye solution, the adsorption was carried out under dark conditions, and the sample was tested every 30 min. After adsorption equilibrium, the solution was placed under 250 w UV lamp (main spectrum 365 nm) for light treatment, and samples were tested every 60 min. The degradation efficiency *η* (%) can be calculated using Equation (4) [32]
(4)$$\eta =(1-\frac{C}{C_{0}})\times 100\%$$
where *C*_0_ and *C* are the solution initial and different periods a concentrations of the dye, respectively.

To evaluate the reusability of the composite membrane, after the reaction was completed, the membrane was regenerated after washing with ethanol and water and then immersed in another fresh dye solution for the next photocatalytic experiment under the same conditions.

### 2.6. Characterization

The crystalline structure of TiO_2_ NPs on membranes was analyzed by X-ray diffractometer (XRD, Bruker D8 Advance, Billerica, MA, USA) at the scanning range of 10–80°. Attenuated total reflection Fourier transform infrared spectra (ATR-FTIR, Nicolet 5700, Thermo Scientific, Madison, WI, USA) and X-ray photoelectron spectra (XPS, Kratos XSAM800, Nanuet, NY, USA) were used to evaluate the chemical composition of membranes. The surface morphology and elements of membranes was observed by field emission scanning electron microscopy (FESEM, Hitachi S-4800, Tokyo, Japan). The mechanical properties of samples were measured by a tensile testing machine. The optical property of membranes was evaluated by UV-Vis diffuse reflectance spectroscopy (DRS, HACH DR3900, Loveland, CO, USA) in the range of 320–1100 nm.

## 3. Results and Discussion

### 3.1. XRD Analysis

Figure 3 shows the XRD of anatase TiO_2_, PVDF, M5, and M5-TiO_2_ (2) to reveal the crystalline structure of the resulting materials. A diffraction peaks at 2θ = 20.6° were found, which constituted the typical XRD patterns of PVDF [29]. As for M5-TiO_2_ (2), other diffraction peaks at 2θ = 25.49°, 38.03°, 48.13°, 54.27°, 55.22°, 62.95°, 68.57°, 70.23°, and 75.51° were attributed to (101), (004), (200), (105), (211), (204), (116), (220), and (215) planes of anatase TiO_2_ [33], meanwhile, the diffraction peaks of brookite and rutile TiO_2_ no found [34]. The results show that the presence of the carboxyl group can effectively control the crystal form of TiO_2_ to be anatase type.

### 3.2. The Compositions of Membranes

The functional groups of the PVDF, PVDF/PEMA (25%), M5, and M5-TiO_2_ (2) membrane were analyzed by FTIR spectra, and the results are shown in Figure 4a. It can be found from the figure that all the membranes exhibited peaks at 874 cm^−1^, 1175 cm^−1^, and 1402 cm^−1^, which were caused by the stretching vibration of C-C, -CF_2_-, -CH_2_-, respectively. As for PVDF/PEMA (25%), three characteristic absorption peaks at 1710 cm^−1^, 1840 cm^−1^, and 1780 cm^−1^ were caused by the stretching vibration of the C = O and C-O-C absorption peaks of maleic anhydride. M5 was prepared from PVDF/PEMA (25%) membrane after acid treatment, the absorption peak intensity at 1840 cm^−1^ and 1780 cm^−1^ almost disappeared, while the absorption peak intensity at 1710 cm^−1^ became stronger, suggesting the anhydride was converted to carboxyl groups. In addition, the M5-TiO_2_ (2) membrane displayed a new broad characteristic absorption peak at 400–900 cm^−1^, which was caused by the stretching vibration of O-Ti-O bonds [35]. Simultaneously, the absorption peak intensity at 2650–3600 cm^−1^ became stronger, which was caused by the stretching vibration of -OH. This result indicated that composite membranes loading with TiO_2_ were successfully prepared.

In order to verify whether the prepared membrane is photoresponsive and the wavelength of light response, we analyzed the optical properties of the membrane by using UV-vis DRS, and the results are shown in Figure 4b. It can be seen that M5-TiO_2_ (2) membrane has a stronger absorption peak at 262–378 nm, which is mainly due to the efficient UV harvesting of TiO_2_. Meanwhile, no new absorption peaks were found in the whole test band for PVDF and M5 membrane. The results show that the composite membrane has a high efficiency of light energy utilization, which is beneficial to the formation of photogenerated pores on the catalyst on the membrane, thereby improving the photocatalytic efficiency [36,37].

The surface chemical composition of the membrane was analyzed by XPS. Figure 5a shows the full XPS spectra of PVDF, M5, and M5-TiO_2_ (2). It can be seen that all the membranes have a C 1 s peak at 285.46 eV and an F 1s peak at 687.37 eV. As for M5 and M5-TiO_2_, the O 1 s peak at 531.97 eV appeared. When TiO_2_ was deposited on the composite membrane surface, the level peak of Ti 2p appeared, which confirmed the presence of the Ti element. The chemical element composition and content of the membrane are shown in Table 1, and the results are consistent with the above XPS spectra.

Meanwhile, the two peaks of Ti 2p_3/2_ and Ti 2p_1/2_ visible in the high-resolution XPS spectra (Figure 5b) at 459.12eV and 464.5eV, respectively, and the splitting energy between Ti 2p_3/2_ and Ti 2p_1/2_ is about 5.38 eV, indicating the existence of normality for Ti^4+^, similar to the reported data for TiO_2_ [38]. At the same time, the other two peaks appeared at 462.75 (Ti 2p_1/2_) and 457.2 eV (Ti 2p_3/2_) confirming the presence of Ti^3+^ in the titania lattice [39]. In addition, as shown in Figure 5c,d, by comparing the high-resolution spectrum of O1s of M5 and M5-TiO_2_ (2) can be found two new peaks from left to right (the new binding energy of M5-TiO_2_ (2) are 531.10 eV and 529.20 eV), which can be attributed to the O_v_ and the Ti-O on the structure [27]. The results suggest that such a carboxyl induction strategy can create Ti^3+^ defects in TiO_2_ lattices. We think that under high-temperature conditions, the electrons of the carboxyl group will break the Ti-O bond, forming Ov and Ti^3+^ in TiO_2_ [40].

### 3.3. Membrane Morphologies

The surface morphology of fibrous membrane were investigated using SEM. As shown in Figure 6, all membranes consist of randomly oriented nanofibers, forming a massive porous structure, which facilitates adequate contact between dyes and active sites [41]. Figure 6a shows that the fibers of PVDF membrane have a smooth surface and different fiber diameters. For composite membranes prepared by adding PEMA, the fiber diameter of the composite membrane became larger and more uniform, and the grooves appeared on the surface of fiber (Figure 6b). At the same time, the fiber morphology of the composite membrane basically did not change after acid treatment (Figure 6c). In addition, with the increase of PEMA content, the fiber diameter gradually increases, and the fiber surface grooves are more obvious (Appendix A). Increased fiber size and surface roughness lead to the increased mechanical strength of the membrane (Appendix A). When the PEMA content is too high (25%) the cracks are found on the fiber surface (Appendix A), resulting in a reduction in the mechanical strength of the composite membrane (Appendix A). The main reasons are the difference in surface tension between PVDF and PEMA and the formation of the Taylor cone at the needle tip during spinning, which leads to obvious grooves on the surface of blended fibers, and the grooves on the fiber surface are more obvious at high PEMA content. After in situ deposition of TiO_2_, a large number of micro-nano particles appeared on the surface of the membrane fibers(Figure 6d). With the increase in deposition time, the content of micro-nano particles on the membrane surface gradually increased, and the phenomenon of pore blocking occurred when the time was too long (Appendix A).

In order to investigate the distribution of TiO_2_ in the composite membrane, the distribution of Ti and O elements on the surface and section of M5-TiO_2_ (2) membrane was tested by EDS, and the results are shown in Figure 7. It can be seen that Ti and O elements are evenly distributed not only on the surface of the composite membrane but also on the inside of the membrane. This uniform dispersion ensures that the composite membrane has good hydrophilic and photocatalytic properties.

### 3.4. Hydrophilicity of Membranes

To explore the effects of PEMA content and TiO_2_ content on the hydrophilicity of composite membranes, the wettability and permeability of composite membranes were analyzed by measuring the surface water contact angle (WCA) and penetration water flux of the membranes. As shown in Figure 8a, with the increase of PEMA content, the WCA of the composite membrane first decreased and then increased, and the penetration flux first increased and then decreased. When the content of PEMA was 25%, the hydrophilicity of the composite membrane was the best, the WCA was 61.16°, and the pure water permeation flux was 318.47 L·m^−2^·h^−1^ under 0.1 bar transmembrane pressure. The main reason is that with the increase of PEMA content, the number of hydrophilic groups on the membrane surface gradually increases, which makes the membrane has a better binding ability with water. When the content of PEMA is too high, it may be because the split fibers make PVDF more exposed and increase the surface hydrophobic group content. Therefore, according to the hydrophilic properties (Figure 8a) and tensile strength (Appendix A) of the composite membrane, the M5 has the best performance, and the subsequent deposition experiment of TiO_2_ is conducted in the M5 membrane.

Figure 8b investigates the influence of TiO_2_ deposition time on the hydrophilicity of the composite membrane, it can be found that the WCA decreases first and then increases with the increase of TiO_2_ deposition time. With the increase of the deposition time from 0 to 2 h, the values of WCA decrease from 61.16° (M5 to 15.63° (M5-TiO_2_ (2)), and the values of *J_w_* increase from 318.47 (M0) to 5547.57 L·m^−2^·h^−1^ (M5) under 0.1 bar transmembrane pressure. Such excellent superhydrophilicity of M5-TiO_2_ (2) membrane may be derived from the following key factor: (i) the appearance of TiO_2_ not only increases the number of hydrophilic groups in the membrane, but also the micro-nano particles make the surface of the fiber rougher; (ii) the deposition of TiO_2_ is carried out under strong acid and high temperature, the maleic anhydride bond internal of the composite membrane fiber will be further hydrolyzed with the progress of the deposition time, which will increase the overall carboxyl group content of the fiber, resulting in a decrease of WCA and an increase of *J_w_*. When the content of micro-nano particles is too large, the micropore on the membrane surface is blocked (Appendix A), and the surface roughness of the membrane will be reduced, which will increase the membrane contact angle and reduce the permeation flux.

### 3.5. Mechanical Properties of Membrane

The excellent mechanical properties of the membrane are crucial for their practical applications. Evaluation of the mechanical properties of membranes by testing stress-strain curves. As shown in Figure 9, it can be seen that the mechanical properties of the composite membrane are better than PVDF membrane(M0), both tensile strength (from 4.75 Mpa to 9.84 Mpa) and tensile strain (from 47.37% to 130%) increased with increasing PEMA content from 0 wt% to 25 wt%. On the one hand, the composite membrane contains a large number of -COOH and -COO^−^; these groups can not only form hydrogen bonds with each other but also form hydrogen bonds with F in the PVDF chain, thus enhancing the mechanical properties of the composite membrane. On the other hand, when blending PEMA to prepare the composite membrane, not only does the fiber diameter increase but also the grooves appear on the fiber surface, and increased fiber size and surface roughness lead to the increased mechanical strength of the membrane. After in situ deposition of TiO_2_, the tensile strength further increases (from 9.84 Mpa to 10.60 Mpa) and the tensile strain decreases (from 130% to 91.86%), mainly due to the further increase in the surface roughness of the composite membrane. In conclusion, the prepared bifunctional composite membranes have excellent mechanical properties and have potential application value in wastewater treatment.

Since the M5-TiO_2_ (2) membrane showed excellent hydrophilicity and mechanical property, all the subsequent experiments were carried out using this composite membrane.

### 3.6. Separation Performance of the Membrane

In the process of oil/water separation, the separation performance of the membrane is determined by whether the membrane has excellent superoleophobicity underwater, which is usually verified by underwater oil contact angle (UOCA) and anti-oil droplet adhesion experiments. Dichloromethane was selected as the experimental oil, and the prepared composite membrane was tested for underwater oil droplet adhesion resistance and UOCA. The results are shown in Figure 10. It can be seen that when the composite membrane is completely in water, dichloromethane droplets fully contact the surface of the membrane and then lift up and no oil droplets remain on the surface of the membrane, which indicates that the prepared composite membrane has excellent underwater oil adhesion resistance. At the same time, the contact angle of dichloromethane underwater was tested, and the UOCA was as high as 156.65°, indicating that the prepared composite membrane has underwater superoilphobicity.

In order to study the oil/water separation performance of the composite membrane, the as-prepared membrane was placed at the junction of the filtration device, and then the soybean oil-in-water emulsion was poured into the filtration device, and the separation experiment was carried out at the transmembrane pressure of 0.1bar. As shown in Figure 11a, it was found that the M5-TiO_2_(2) membrane had high separation efficiency (98.95%) and penetration flux (678.64 L·m^−2^·h^−1^) for soybean oil-in-water emulsion. Meanwhile, the FRR of M5-TiO_2_ (2) membrane reached 99.19%, which was much higher than that of the M0 and M5 membranes (Figure 11b), and the M5-TiO_2_ (2) membrane has good permeability and separation efficiency compared with other membrane materials (Appendix A) [42,43,44]. Meanwhile, optical microscope images and particle size distribution before and after emulsion separation were tested, and the results are shown in Figure 11c. It can be seen that the emulsion before separation is milky white and cloudy. The particle size of oil droplets in the emulsion is mainly concentrated in 100–1000 nm, and there are some extreme values at the edge. The filtrate after membrane separation is relatively clear, and the particle size in the filtrate is mainly concentrated in the hundreds of grades. We believe that the appearance of the particle size of oil droplets in the filtrate is mainly caused by Tween 80. The reusable performance of the membrane is also one of the important properties of the separation membrane. In order to further verify the reusable performance of the membrane, five consecutive oil–water separation experiments were conducted on the membrane, and the results were shown in Figure 11d. It is worth noting that after each separation experiment, the membrane was cleaned with deionized water for 5 min before the experiment was conducted again. As can be seen from Figure 11d, the separation efficiency of M5-TiO_2_ (2) membrane in 5 separation experiments all reached more than 98%, and had a high permeability flux. The results show that the composite membrane has good recyclability and pollution resistance.

Such excellent superoleophobicity and anti-oil-adhesion of the composite membrane under water may originate from the following key factors: (i) -COOH and -COO^−^ have high water molecule binding ability, and through the strong hydrogen bonding between -COOH/-COO^−^ and water molecules, it exhibits excellent water absorption and water retention capacity, forming a highly stable hydration sheath on the membrane surface, according to the classical Cassie model (oil/water/solid three-phase interface) [45,46], the dense hydration layer acts as a strong barrier, which can effectively prevent oil droplets from directly contacting and adhering to the membrane surface, thereby greatly reducing the contact area between oil and the membrane surface, thereby achieving excellent oil-repellent performance. (ii) The rough mesh forms many nanocavities, providing more space to trap water molecules, which apparently promotes the formation of the Cassie state.

### 3.7. Separation Mechanism of Oil–Water Emulsion

As shown in Figure 12, the M5-TiO_2_ (2) membrane exhibits superhydrophilicity and underwater superoleophobicity due to its high surface energy and special micro-nanostructure. When water comes into contact with the membrane, the water molecules will rapidly hydrogen-bond with the hydrophilic groups on the membrane surface. At the same time, due to the existence of the micro-nano structure, more water molecules are trapped on the surface to form a hydration layer, thus forming a strong repelling barrier to resist the contamination of oil droplets. In addition, capillary action can further reveal the superhydrophilic and underwater superoleophobic properties of M5-TiO_2_ (2) membrane. As shown in Appendix A, the mean pore size of the M5-TiO_2_ (2) membrane is 1.75 μm, which conforms to capillary mechanics, and these pores can be referred to as abundant capillary tubes. According to the Jurin formula [47], the Jurin height of the M5-TiO_2_(2) membrane is calculated to be about 16.2 m, which is much higher than the thickness of the M5-TiO_2_ (2) membrane (0.2 mm). Consequently, the water molecules can be achieved on the superhydrophilic membrane surface and maintain continuous penetration under capillary action. Even though oil drops can access the pores of the superhydrophilic membrane, they are pushed back due to the upward pressure from capillary action. Therefore, the M5-TiO_2_ (2) membrane presented a high-efficiency separation performance for oil–water emulsions.

### 3.8. Photocatalytic Degradation of Organic Dyes

The photocatalytic activity of the composite membranes was analyzed by RhB (cationic dye, initial concentration 5mg/L, pH = 7) and AG-25 (anionic dye, initial concentration 50 mg/L, pH = 3) in the dark and under UV irradiation, respectively, and the results were shown in Figure 13. It can be found from Figure 13(a1,b1) that RhB and AG-25 concentrations decreased slightly during the first 3 h under dark conditions. By adjusting pH, the composite membrane can have a certain adsorption on cationic dyes or anionic dyes, the main reasons are as follows: carboxylic acid pKa = 4 [48], when pH = 7, most of the carboxyl groups on the membrane M5-TiO_2_ (2) surface will be hydrolyzed into carboxyl ion. Meanwhile, in such conditions, the quaternary ammonium groups in RhB are in cationic form, which promotes ionic polymerization between quaternary ammonium and the carboxylic anionic group [49]. For AG-25 dye, when pH = 3, most of the carboxyl groups on the membrane M5-TiO_2_ (2) surface will exist in the protonated form. Meanwhile, in such conditions, the sulfonic acid group in AG-25 is in the form of an anion due to the pKa = 2.8 of the sulfonic acid [50], which promotes the ionic dipole interaction between the sulfonic acid anion and the carboxyl group [49]. After adsorption saturation, for the dye solutions containing membrane M5, the concentrations of RhB and AG-25 did not change substantially under UV irradiation. For dye solutions containing membrane M5-TiO_2_ (2), the concentrations of RhB and AG-25 decreased rapidly under UV irradiation, reaching 97% and 90.2% removal rates, respectively. The changes of UV-vis light absorbance of RhB and AG-25 solution at different times are shown in Figure 13(a2,b2). It can be seen from the figure that RhB solution shows strong absorbance at 554 nm, and the corresponding absorbance at 554 nm almost disappears after adsorption and degradation. AG-25 solution showed strong absorbance at 608 nm and almost disappeared after degradation by adsorption. The main reason is that the conjugated chromophore structure of the dye is destroyed in the photocatalytic degradation process, and the dye molecules are completely degraded into small organic/inorganic molecules or/and ionic products.

Meanwhile, the M5-TiO_2_ (2) membrane prepared in this study has excellent photocatalytic efficiency and high photocatalytic degradation efficiency for both anionic and cationic dyes compared with the membrane materials prepared in other studies (Appendix A) [42,43,44]. The reusability of photocatalytic membranes is also one of the important properties used to evaluate the membrane in practical applications [51]. We conducted an experiment on the reusability of the prepared composite membrane, and the results are shown in Appendix A. After each photocatalytic dye degradation experiment, the M5-TiO_2_ (2) membrane was cleaned with deionized water and then placed in fresh dye solution for repeated adsorption degradation experiments. It can be seen from Appendix A that the degradation efficiency of the M5-TiO_2_ (2) decreases slightly after running for five times, but the M5-TiO_2_ (2) exhibited great recyclability with 96% and 88% removal ratio for RhB and AG-25 dye, demonstrating that the photocatalytic performance of the M5-TiO_2_ (2) is relatively stable and can be recycled. At the same time, the TiO_2_ supported on the membrane effectively overcomes the problem that the powder photocatalyst is difficult to recycle. Therefore, this composite membrane is a green material, and it is easier to realize recycling in practical use.

### 3.9. Mechanism of Photocatalysis of Membrane

The process of dye removal by M5-TiO_2_ (2) membrane consisted of two main steps: one was dye adsorption, the other was dye photocatalytic degradation. The former could be achieved through the porous structure of the composite membrane, -COOH/-COO^−^ and TiO_2_ nanoparticles on the membrane surface. The latter was depicted in Figure 14, anatase TiO_2_ has been shown by numerous studies to be more photocatalytically active due to having lower surface energy than rutile. Under UV irradiation, the valence band electrons of anatase TiO_2_ were excited to the conduction band (CB), the photogenerated electrons on the CB reacted with O_2_ to form ·O_2_^−^. The introduction of Ti^3+^ and O_v_ can produce an impurity level just below the CB, the electrons trapped by O_v_ and hopping between Ti^3+^ and Ti^4+^, which is helpful to improve the concentration of photogenerated carriers and utilization rate of light. On the other hand, the photogenerated holes on the valence band (VB) reacted with H_2_O or OH^−^ to form ·OH. Under the action of ·O_2_^−^, ·OH, and the photogenerated holes, the organic dye can be effectively degraded into CO_2_, H_2_O, and other degradation products [52].

## 4. Conclusions

In this work, bifunctional PVDF composite membranes with excellent oil-in-water emulsion separation and photocatalytic degradation properties were successfully prepared by electrostatic spinning technology and hydrothermal reaction method. A large number of -COOH and anatase TiO_2_ with Ti^3+^ and O_v_ particles on the membrane surface make the composite membrane have strong hydrophilic properties (WCA = 15.65), underwater oleophobic properties (UOCA = 156.69) and photocatalytic performance. These composite membranes have high separation efficiency and outstanding anti-fouling performance, the oil removal efficiency reaches 98.95%, and the FRR reaches 99.19% for soybean oil-in-water emulsion. In addition, the composite membrane has excellent photocatalytic degradation performance, with 97% and 90.2% degradation of RhB and AG-25 under UV conditions, respectively. In summary, the bifunctional PVDF composite membrane provides a new idea for the simultaneous removal of oil and dyes in wastewater, and has broad application prospects in water-environment remediation.

## Figures and Tables

**Figure 1 membranes-13-00364-f001:**
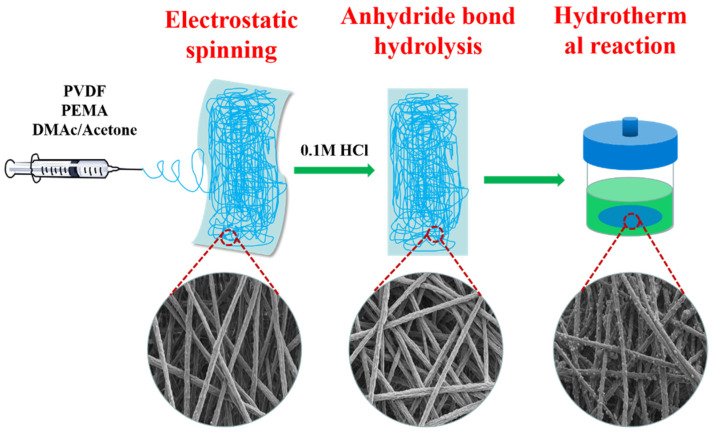
The preparation process of the bifunctional composite membrane.

**Figure 2 membranes-13-00364-f002:**
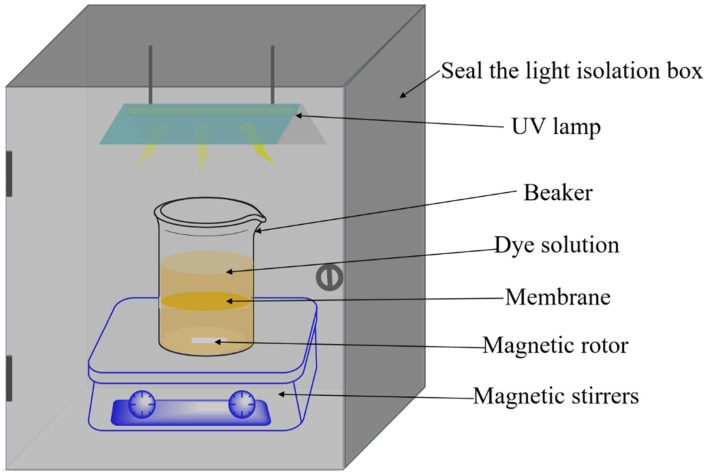
Schematic of the photoreactor equipment.

**Figure 3 membranes-13-00364-f003:**
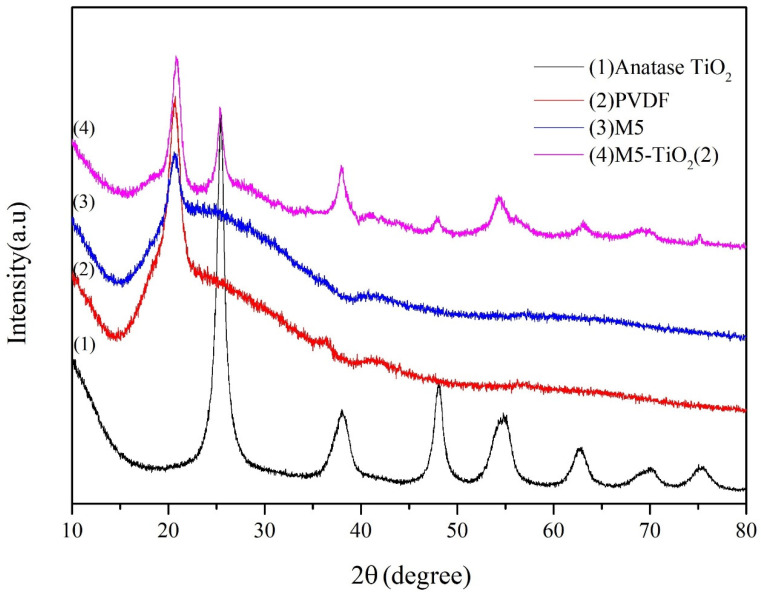
XRD patterns of anatase TiO_2_, PVDF, M5, and M5-TiO_2_ (2).

**Figure 4 membranes-13-00364-f004:**
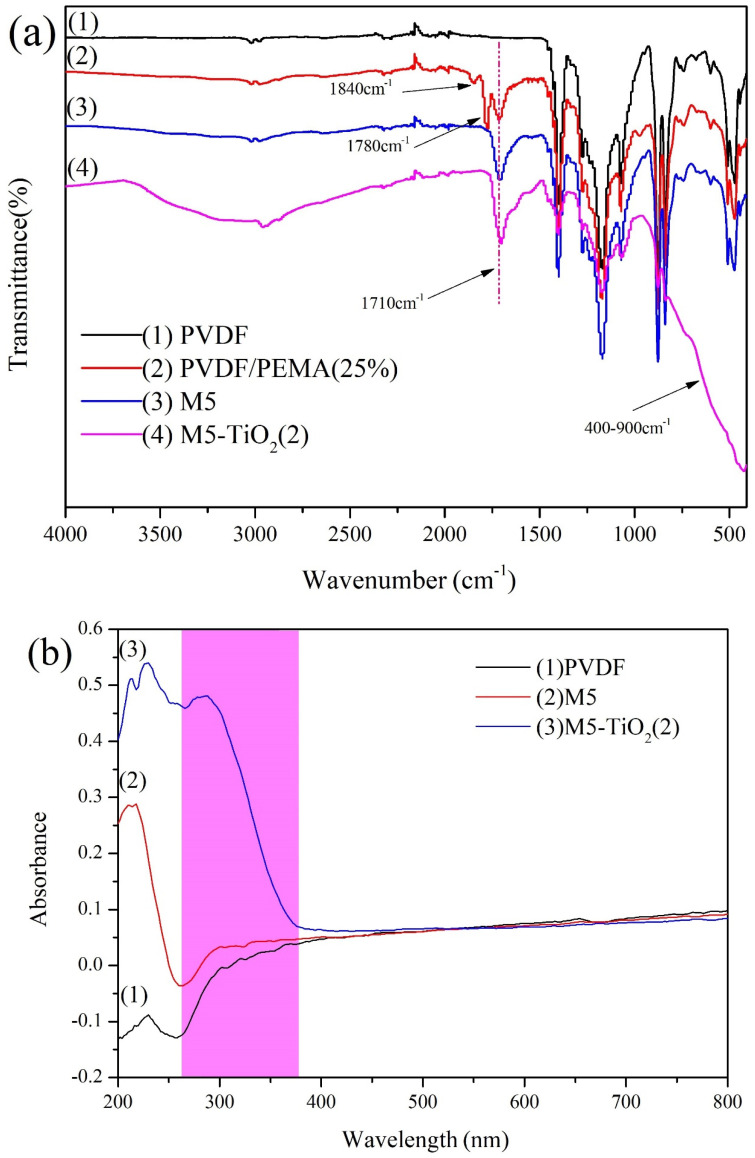
(**a**) FTIR spectra of the PVDF, PVDF/PEMA (25%), M5 and M5-TiO2(2), (**b**) UV-vis DRS spectrum of the membranes.

**Figure 5 membranes-13-00364-f005:**
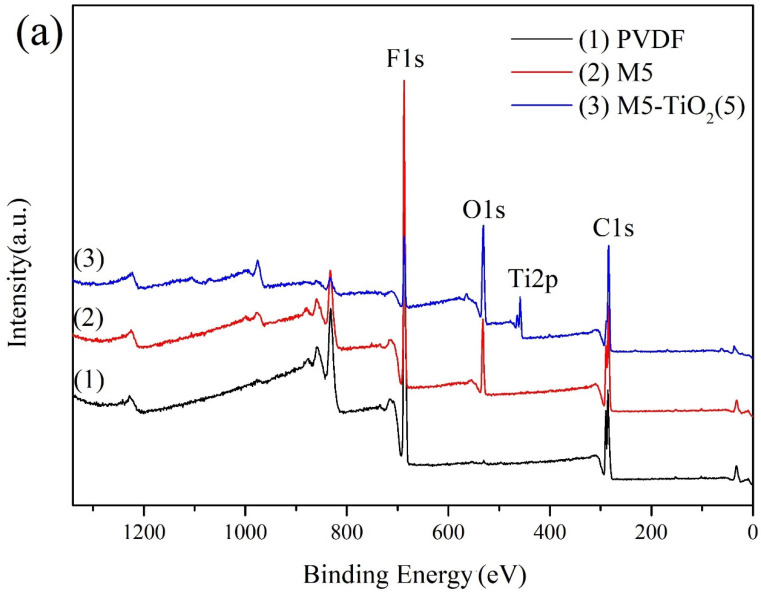
Survey XPS spectra (**a**), Ti 2p fitting curves of M5-TiO_2_(2) (**b**), O1s fitting curves of M5 (**c**) and O1s fitting curves of M5-TiO_2_ (2) (**d**).

**Figure 6 membranes-13-00364-f006:**
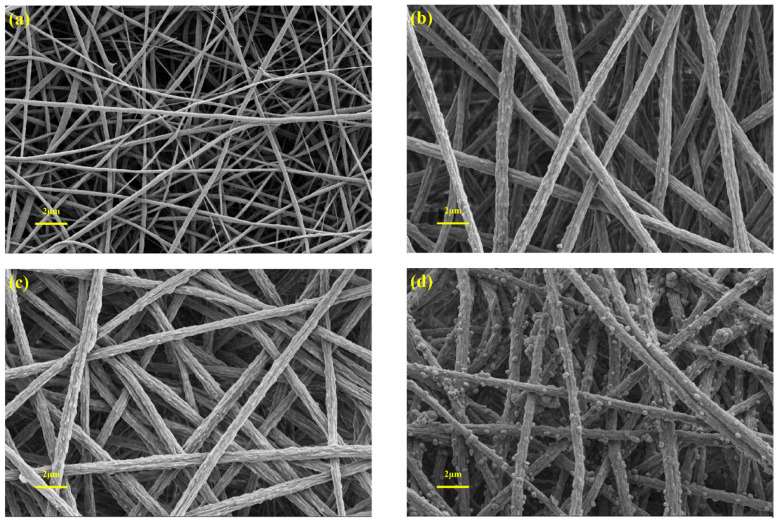
SEM images of PVDF (**a**), PVDF/PEMA (25%) (**b**), M5 (**c**), and M5-TiO_2_ (2) (**d**).

**Figure 7 membranes-13-00364-f007:**
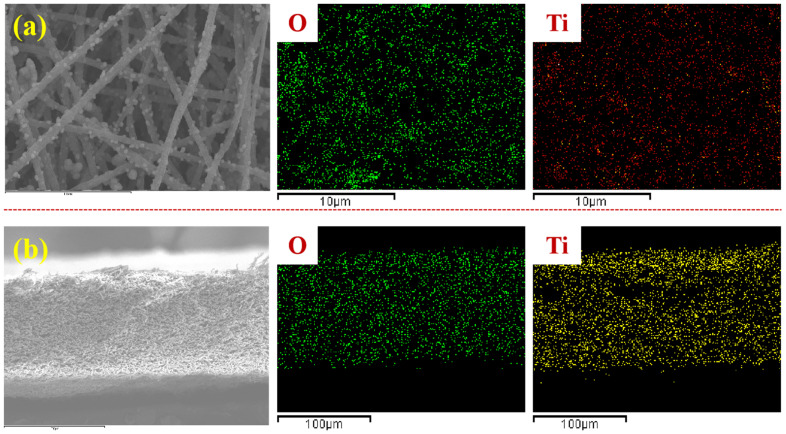
The EDS mapping images of M5-TiO_2_ (2) surface (**a**) and cross-section (**b**).

**Figure 8 membranes-13-00364-f008:**
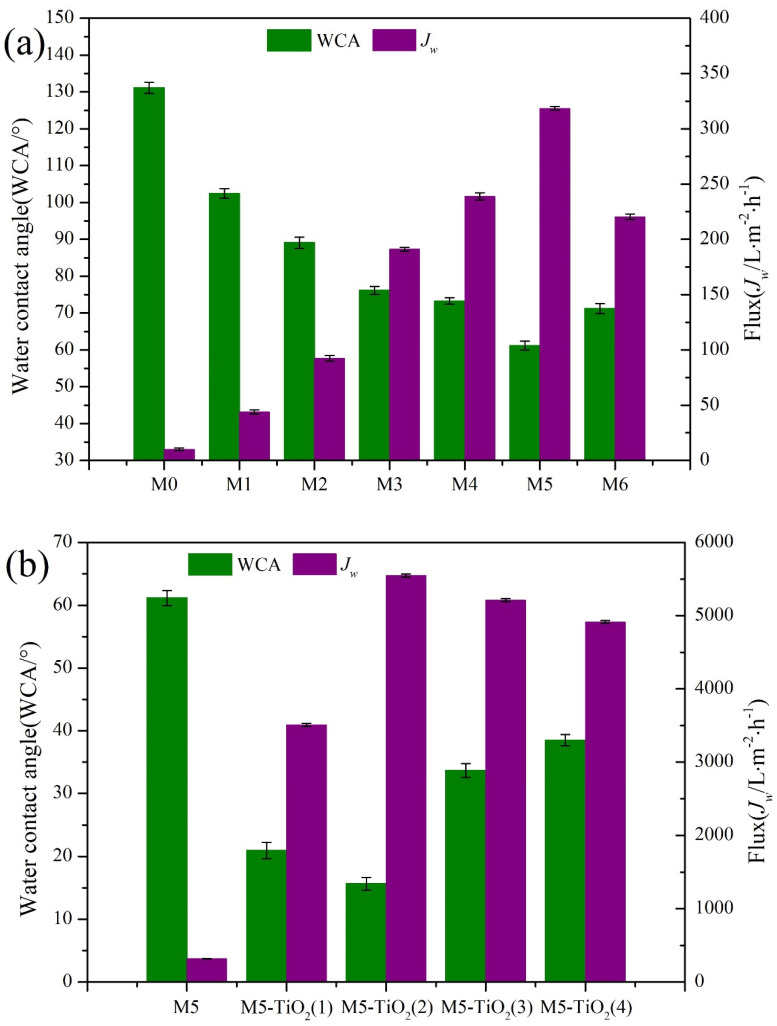
The affection of PEMA content (**a**) and TiO_2_ deposition time (**b**) on the hydrophilicity of membranes.

**Figure 9 membranes-13-00364-f009:**
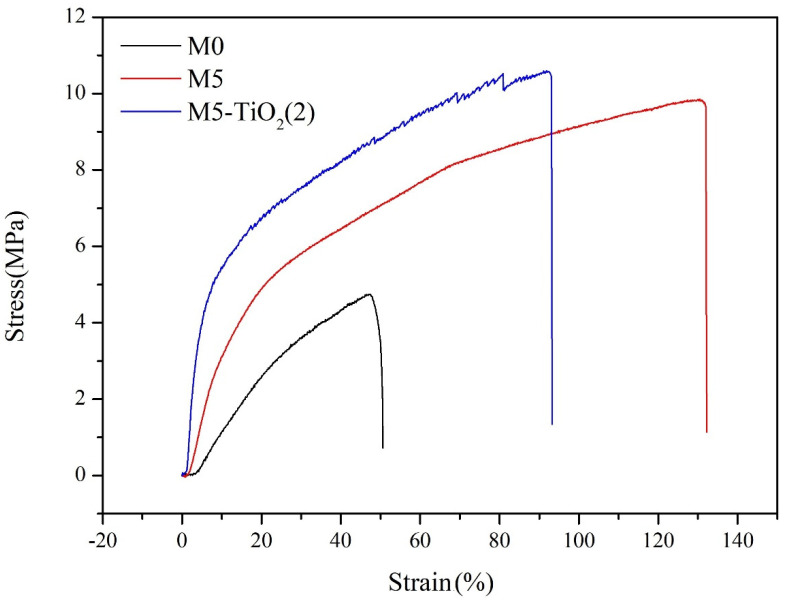
The stress-strain curves of the membranes.

**Figure 10 membranes-13-00364-f010:**
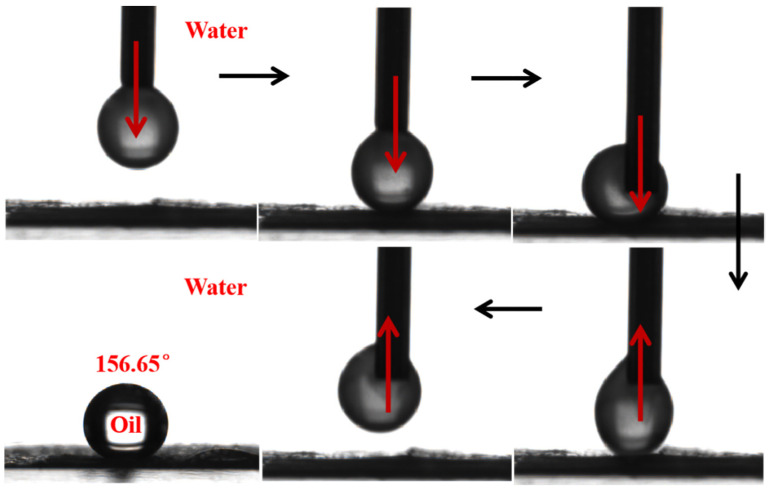
Dynamic underwater oil adhesion and underwater oil contact angle of the M5-TiO_2_ (2) membrane (dichloromethane).

**Figure 11 membranes-13-00364-f011:**
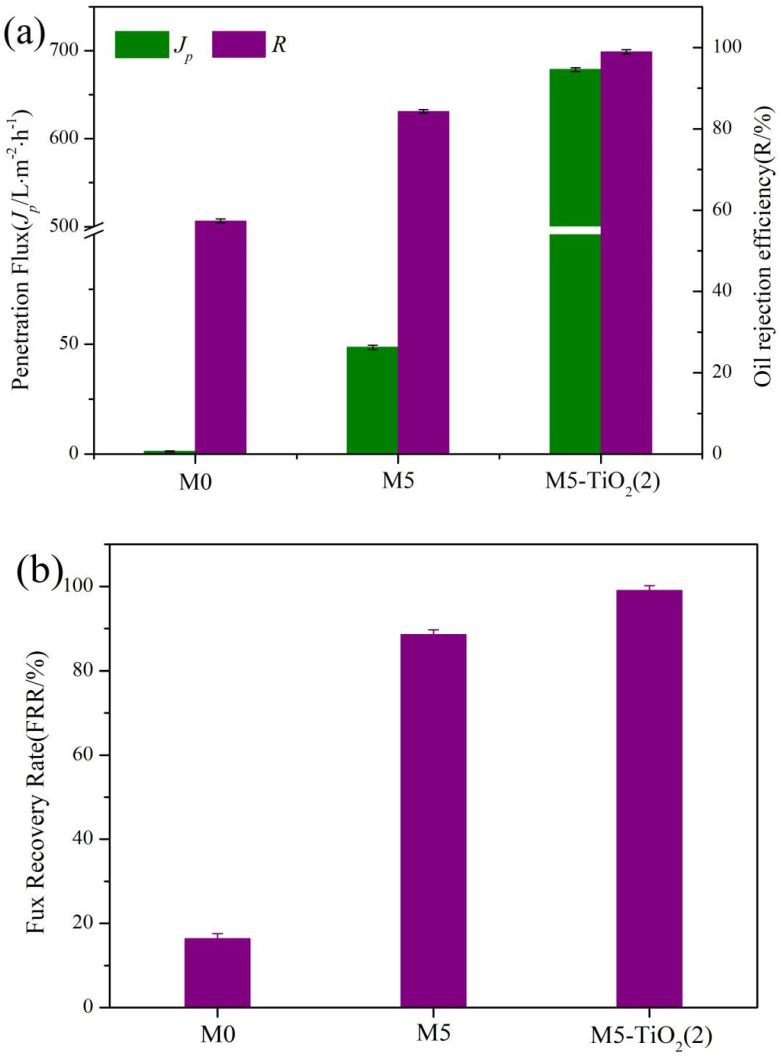
(**a**) Separation performance of membranes, (**b**) flux recovery rate of membranes, (**c**) optical micrographs and particle size distribution of emulsions before and after separation, and (**d**) recyclability separation experiment of emulsion.

**Figure 12 membranes-13-00364-f012:**
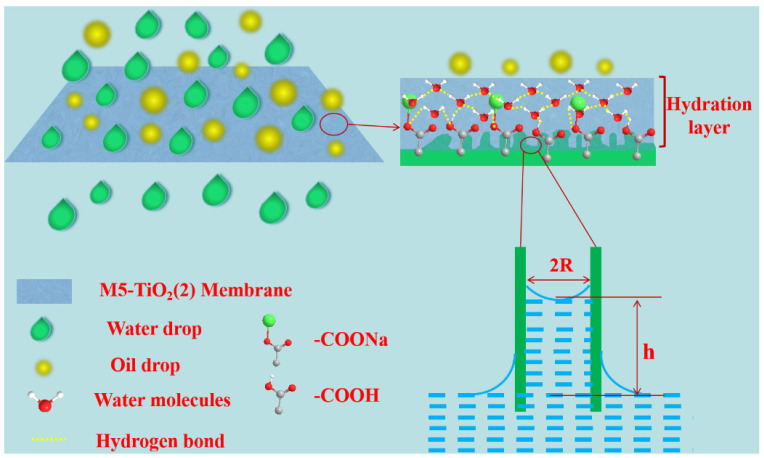
Mechanism of separation of oil-in-water emulsion by M5-TiO_2_ (2) membrane.

**Figure 13 membranes-13-00364-f013:**
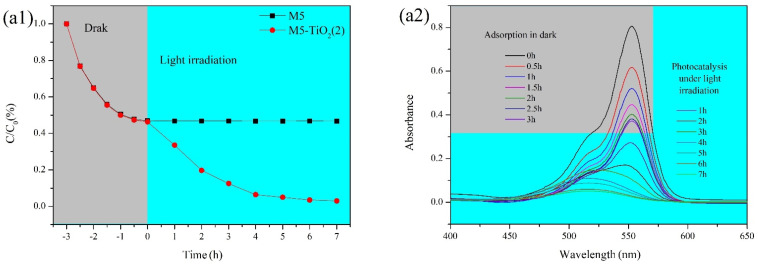
Photocatalytic activity under UV conditions for the degradation of organic dyes:. degradation efficiency of RhB (**a1**) and AG-25 (**b1**); absorbance at different times (RhB (**a2**), AG-25 (**b2**)).

**Figure 14 membranes-13-00364-f014:**
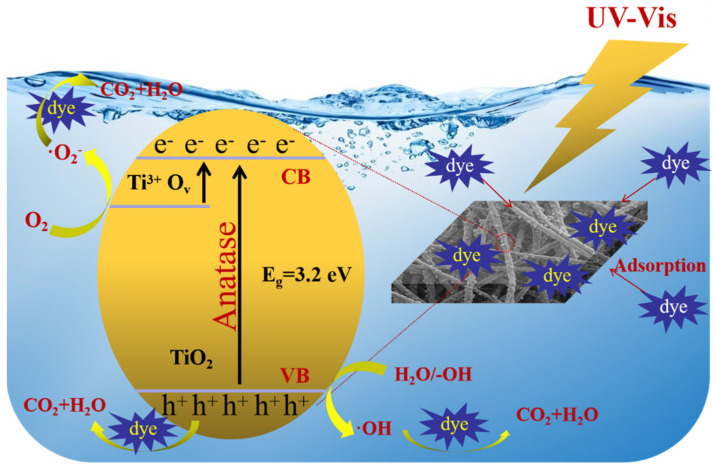
The proposed mechanism of photodegrading dyes in the presence of M5-TiO_2_(2) under UV irradiation.

**Table 1 membranes-13-00364-t001:** Elemental composition of different membranes as determined by XPS.

Membrane	Composition (at%)			
	C	F	O	Ti
PVDF	51.23	48.77	/	/
M5	54.7	34.29	11.01	/
M5-TiO_2_ (2)	59.28	12.21	24.15	4.36

## Data Availability

Not applicable.

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
