# Peer review of "Fabrication of Anatase TiO2/PVDF Composite Membrane for Oil-in-Water Emulsion Separation and Dye Photocatalytic Degradation"

_membranes, 2023, doi:10.3390/membranes13030364_

Round 1

Reviewer 1 Report

The work entitled “Fabrication of anatase TiO2/PVDF composite membrane for oil-in-water emulsion separation and dye photocatalytic degradation” is an interesting study on the development of composite membranes for water treatment. The work is well-written and organized. Some specific comments:

1) Abbreviations should be explained before used. Please check throughout the manuscript. Example: FRR was used in the Abstract section without being explained.

2) Keywords must be improved. Some keywords are too long and seem like phrases. Some keywords are already present in the title, which is also not good.

3) Lines 39-43: There is no better way to use a photocatalyst. Photocatalytic membranes are just a great approach. Here, I suggest authors compare the use of photocatalysts in suspension or immobilized on the membrane. Please check: 10.1016/j.cej.2019.122114.

4) How the concentration of RhB and AG-25 was measured?

5) Why did the authors use such a high concentration?

6) I suggest that the authors add a schematic of the photoreactor used.

Author Response

1) Abbreviations should be explained before used. Please check throughout the manuscript. Example: FRR was used in the Abstract section without being explained.

Answer:As our carelessness leads to the emergence of this problem, it is deeply guilty and has been modified. The abbreviation for flux recovery rate is FRR.

2) Keywords must be improved. Some keywords are too long and seem like phrases. Some keywords are already present in the title, which is also not good.

Answer:As our carelessness leads to the emergence of this problem, it is deeply guilty and has been modified. The Keyword have been replaced with Membrane, TiO2, Emulsion separation and Photocatalytic degradation.

3) Lines 39-43: There is no better way to use a photocatalyst. Photocatalytic membranes are just a great approach. Here, I suggest authors compare the use of photocatalysts in suspension or immobilized on the membrane. Please check: 10.1016/j.cej.2019.122114.

Answer:We will be happy to edit the text further, based on helpful comments from the reviewers. However, we have not understood the specific content to be compared, and the article with this link(10.1016/j.cej.2019.122114) has not been found. We hope you can send us a new link, and we will revise it after reading it carefully.

4) How the concentration of RhB and AG-25 was measured?

Answer:We are sorry for the lack of explanation in the text. First, the standard curve of dye absorbance and concentration is tested, and then the concentration is calculated according to the value of dye absorbance during photocatalytic degradation.

5) Why did the authors use such a high concentration?

Answer:We are sorry for the lack of explanation in the text. For RhB, in order to compare its catalytic performance with that in other literatures, the same concentration was selected as that in the literatures. For AG-25, we found in the test that it was difficult to test the corresponding absorbance when AG-25 concentration was too low(<5mg/L), which had a certain impact on the photocatalytic results.

6) I suggest that the authors add a schematic of the photoreactor used.

Answer:As our carelessness leads to the emergence of this problem, it is deeply guilty and has been modified.

Reviewer 2 Report

This work describes the preparation of hydrophobic TiO2/PVDF membranes prepared by in situ deposition of anatase. The membranes were evaluated for oil-water separation and finally, the photocatalytic activity for dye removal was explored. The work is interesting however, several modifications are required before publication.

The novelty of the work should be better explained in the introduction section.

The rationale structure of the study is not clear in the manuscript. The discussion section should be improved to explain some experiments instead of only presenting characterization results without context.

Line 71-71: The authors express that anatase “has the highest photocatalytic activity and is the most widely used” however, the benchmark TiO2 catalyst is a mixture of anatase rutile.

Section 2.5: the wavelength of the UV lamp is not detailed.

Section 2.6: Authors should give details regarding each equipment used for characterization.

All Figures has poor quality which makes it difficult the interpretation. Figure 12 is not possible to analyse.

Figure 3a there is written SiO2 instead of TiO2.

Line 203-206: The FTIR spectra of PVDF/PEMA(25%) after acid treatment are not presented in the figure. It should be demonstrated.

Minor correction:

Line 131: “dripped into 30mL of 2mol/L H2SO4”

Line 141: Or present “delta t” with a capital letter or correct Equation 1.

Sometimes the authors write FT-IR and other FTIR. Please uniformize.

Line 213: “TiO2”

Line 21/23 (SM): “form” should be from

Figure S4 caption: Membrane; Authors should detail de deposition time of each panel to explain the evolution in the surface.

Figure S5 caption: TiO2

Line 327: “form” should be from

Line 386: and water

Lines 383-392: The sentence is too long, is not clear. Please rephrase.

Line 422: activity

Line 465: M5-TiO2(2)

Author Response

This work describes the preparation of hydrophobic TiO2/PVDF membranes prepared by in situ deposition of anatase. The membranes were evaluated for oil-water separation and finally, the photocatalytic activity for dye removal was explored. The work is interesting however, several modifications are required before publication.

  • The novelty of the work should be better explained in the introduction section.

Answer:We are sorry for the lack of explanation in the text, it is deeply guilty and has been modified. The modified parts have been highlighted in red.

“The obtained TiO2/PVDF composite membrane showed several distinct advantages:(1) The membrane achieves remarkable underwater superoleophobicity property; (2) The membrane can effectively separate oil/water emulsion and good antifouling ability; (3) The membrane has high photocatalytic degradation ability for both anionic and cationic dyes.”

  • The rationale structure of the study is not clear in the manuscript. The discussion section should be improved to explain some experiments instead of only presenting characterization results without context.

Answer:As our carelessness leads to the emergence of this problem, it is deeply guilty and has been modified. The modified parts have been highlighted in red.

  • Line 71-71: The authors express that anatase “has the highest photocatalytic activity and is the most widely used” however, the benchmark TiO2 catalyst is a mixture of anatase rutile.

Answer: We will be happy to edit the text further, based on helpful comments from the reviewers. We have read the articles(G.Q. Liu, X.J. Pan, J. Li, C. Li, C.L. Ji, Facile preparation and characterization of anatase TiO2/nanocellulose composite for photocatalytic degradation of methyl orange, J. Chil. Chem. Soc., 25(12) (2021), 101383; S.M. Tichapondwa, J.P. Newman, O. Kubheka, Effect of TiO2 phase on the photocatalytic degradation of methylene blue dye, Phys. Chem. Earth, 118-119 (2020), 102900) found the peaks at 2θ=25.49°, 38.03°, 48.13°, 54.27°, 55.22°, 62.95°, 68.57°, 70.23° and 75.51° were attributed to (101), (004), (200), (105), (211), (204), (116), (220) and (215) planes of anatase TiO2, meanwhile, the diffraction peaks of brookite and rutile TiO2 no found. For M5-TiO2(2) membrane, other diffraction peaks at 2θ=25.49°, 38.03°, 48.13°, 54.27°, 55.22°, 62.95°, 68.57°, 70.23° and 75.51° were found, and the diffraction peaks of brookite and rutile TiO2 no found. The results show that the presence of carboxyl group can effectively control the crystal form of TiO2 to be anatase type.

  • Section 2.5: the wavelength of the UV lamp is not detailed.

Answer: As our carelessness leads to the emergence of this problem, it is deeply guilty and has been modified. The UV lamp irradiance is 250w, main spectrum 365nm.

  • Section 2.6: Authors should give details regarding each equipment used for characterization.

Answer: We are sorry for the lack of explanation in the text. The modified parts have been highlighted in red. The crystalline structure of TiO2 NPs on membranes was analyzed by X-ray diffractometer (XRD, Bruker D8 Advance, Germany) at the scanning range of 10-80°. Attenuated total reflection Fourier transform infrared spectra (ATR-FTIR, Nicolet 5700, America) and X-ray photoelectron spectra (XPS, Kratos XSAM800, America) was used to evaluate the chemical composition of membranes. The surface morphology and elements of membranes was observed by field emission scanning electron microscopy (FESEM, HitachiS-4800, Japan). The mechanical properties of samples were measured by a tensile testing machine. The optical property of membranes was evaluated by UV-Vis diffuse reflectance spectroscopy (DRS, HACH DR3900, America) in the range of 320-1100 nm.

  • All Figures has poor quality which makes it difficult the interpretation. Figure 12 is not possible to analyse.

Answer:As our carelessness leads to the emergence of this problem, it is deeply guilty. The poor quality Figures have been replaced, and the superfluous content in Figure 12 has been removed.

  • Figure 3a there is written SiO2 instead of TiO2.

Answer: As our carelessness leads to the emergence of this problem, it is deeply guilty and has been modified.

  • Line 203-206: The FTIR spectra of PVDF/PEMA(25%) after acid treatment are not presented in the figure. It should be demonstrated.

Answer: We are sorry for the lack of explanation in the text. M5 was prepared from PVDF/PEMA(25%) membrane after acid treatment, the FTIR spectra of M5 has been fully expressed in Figure 4, the absorption peak intensity at 1840cm-1 and 1780cm-1 almost disappeared, while the absorption peak intensity at 1710cm-1 became stronger, suggesting the anhydride was converted to carboxyl groups.

Minor correction:

  • Line 131: “dripped into 30mL of 2mol/L H2SO4”

Answer: As our carelessness leads to the emergence of this problem, it is deeply guilty and has been modified.

  • Line 141: Or present “delta t” with a capital letter or correct Equation 1.

Answer: As our carelessness leads to the emergence of this problem, it is deeply guilty and the Equation 1 been modified.

  • Sometimes the authors write FT-IR and other FTIR. Please uniformize.

Answer: As our carelessness leads to the emergence of this problem, it is deeply guilty and all the FT-IR were changed to FTIR.

  • Line 213: “TiO2”

Answer: As our carelessness leads to the emergence of this problem, it is deeply guilty and has been modified.

  • Line 21/23 (SM): “form” should be from

Answer: As our carelessness leads to the emergence of this problem, it is deeply guilty and has been modified.

  • Figure S4 caption: Membrane; Authors should detail de deposition time of each panel to explain the evolution in the surface.

Answer: As our carelessness leads to the emergence of this problem, it is deeply guilty and has been modified.

  • Figure S5 caption: TiO2

Answer: As our carelessness leads to the emergence of this problem, it is deeply guilty and has been modified.

  • Line 327: “form” should be from

Answer: As our carelessness leads to the emergence of this problem, it is deeply guilty and has been modified.

  • Line 386: and water

Answer: As our carelessness leads to the emergence of this problem, it is deeply guilty and has been modified.

  • Lines 383-392: The sentence is too long, is not clear. Please rephrase.

Answer: As our carelessness leads to the emergence of this problem, it is deeply guilty and has been modified. 

In order to further verify the reusable performance of the membrane, five consecutive oil-water separation experiments were conducted on the membrane, and the results were shown in Fig. 10d. It is worth noting that after each separation experiment, the membrane was cleaned with deionized water for 5min before the experiment was conducted again. As can be seen from Fig. 10d, the separation efficiency of M5-TiO2(2) membrane in 5 separation experiments all reached more than 98%, and had a high permeability flux. The results show that the composite membrane has good recyclability and pollution resistance.

  • Line 422: activity

Answer: As our carelessness leads to the emergence of this problem, it is deeply guilty and has been modified.

  • Line 465: M5-TiO2(2)

Answer: As our carelessness leads to the emergence of this problem, it is deeply guilty and has been modified.

Round 2

Reviewer 1 Report

The authors have improved the manuscript accordingly, thus I suggest it be published.

Reviewer 2 Report

The manuscript quality was improved after the revision, and I recommend it for publication.